# DNN-m6A: A Cross-Species Method for Identifying RNA N6-methyladenosine Sites Based on Deep Neural Network with Multi-Information Fusion

**DOI:** 10.3390/genes12030354

**Published:** 2021-02-28

**Authors:** Lu Zhang, Xinyi Qin, Min Liu, Ziwei Xu, Guangzhong Liu

**Affiliations:** 1College of Information Engineering, Shanghai Maritime University, Shanghai 201306, China; 201940310007@stu.shmtu.edu.cn (L.Z.); 201930310113@stu.shmtu.edu.cn (X.Q.); liumin@shmtu.edu.cn (M.L.); 2Polytech Nantes, Bâtiment Ireste, 44300 Nantes, France; ziwei.xu@etu.univ-nantes.fr

**Keywords:** N6-methyladenosine sites, multi-information fusion, elastic net, deep neural network, Bayesian hyper-parameter optimization

## Abstract

As a prevalent existing post-transcriptional modification of RNA, N6-methyladenosine (m6A) plays a crucial role in various biological processes. To better radically reveal its regulatory mechanism and provide new insights for drug design, the accurate identification of m6A sites in genome-wide is vital. As the traditional experimental methods are time-consuming and cost-prohibitive, it is necessary to design a more efficient computational method to detect the m6A sites. In this study, we propose a novel cross-species computational method DNN-m6A based on the deep neural network (DNN) to identify m6A sites in multiple tissues of human, mouse and rat. Firstly, binary encoding (BE), tri-nucleotide composition (TNC), enhanced nucleic acid composition (ENAC), *K*-spaced nucleotide pair frequencies (KSNPFs), nucleotide chemical property (NCP), pseudo dinucleotide composition (PseDNC), position-specific nucleotide propensity (PSNP) and position-specific dinucleotide propensity (PSDP) are employed to extract RNA sequence features which are subsequently fused to construct the initial feature vector set. Secondly, we use elastic net to eliminate redundant features while building the optimal feature subset. Finally, the hyper-parameters of DNN are tuned with Bayesian hyper-parameter optimization based on the selected feature subset. The five-fold cross-validation test on training datasets show that the proposed DNN-m6A method outperformed the state-of-the-art method for predicting m6A sites, with an accuracy (ACC) of 73.58–83.38% and an area under the curve (AUC) of 81.39–91.04%. Furthermore, the independent datasets achieved an ACC of 72.95–83.04% and an AUC of 80.79–91.09%, which shows an excellent generalization ability of our proposed method.

## 1. Introduction

The post-transcriptional modification of RNA increases the complexity of biological information and the fineness of regulation. Currently, more than 150 kinds of post-transcriptional modification of RNA have been identified, and two thirds of those modifications are methylated [1]. The two most representative types of Methylation modifications are N6-methyladenosine (m6A) [2] and 5-methylcytosine (m5C) [3,4,5,6]. Compared with m5C, m6A is the most abundant internal modification on mRNA in eukaryotes, accounting for about 80% of all the methylation forms. M6A refers to the methylation modification that occurs on the sixth nitrogen atom of adenosine under the action of the methyltransferase complexes (i.e., METTL3, METTLI4, WTAP, etc.). Moreover, m6A methylation is a dynamically reversible process, which is regulated by methyltransferases and demethylases in time and space [7,8,9]. As an important RNA post-transcriptional modification site, m6A exists in a variety of species including viruses, bacteria, plants, and mammals [10]. Studies have shown that m6A plays a regulatory role in almost every stage of mRNA metabolism [11]. Meanwhile, m6A modification participates in the pathogenesis of multiple diseases including cancers. Accumulating evidence shows that, m6A modification is associated with the tumor proliferation, differentiation, tumorigenesis [12], proliferation [13], invasion [12], and metastasis [14] and functions as oncogenes or anti-oncogenes in malignant tumors [15]. Given the importance of N6-methyladenosine, it is very essential to identify the m6A sites accurately, which can contribute to provide new ideas for biomedical research to better explore and elucidate the regulatory mechanisms of m6A. In addition, it also can aid in understanding disease mechanisms and accelerating the development of the new drug.

Traditional approaches for detecting the m6A sites in RNA are roughly divided into three categories: two-dimensional thin layer chromatography [16], high performance liquid chromatography [17], and high-throughput methods including m6A-seq [18] and MeRIPSeq [19]. However, it is a time and labor-consuming work on detecting m6A sites through traditional experimental methods [20]. Therefore, it is requisite to exploit computing method that can accurately, effectively, efficiently identify m6A sites. Recently, researchers have attained a series of valuable progress in predicting m6A sites of RNA based on machine learning (ML). Zhao et al. [21] constructed a human mRNA m6A sites prediction model HMpre, which used the cost-sensitive approach to resolve the imbalance data issues. Chen et al. [22] developed a m6A sites predictor iRNA-PseDNC using pseudo nucleotide composition to extract features, and the 10-fold cross-validation proved that iRNA-PseDNC has better performance than RAM-NPPS. Chen et al. [23] selected ensemble support vector machine as a classifier to construct a prediction model RAM-ESVM. Xing et al. [24] proposed a prediction model RAM-NPPS based on support vector machine, which used position-specific condition propensity to extract RNA sequence features. Wei et al. [25] generated feature vectors using position-specific nucleotide propensity, physical–chemical properties and ring-function-hydrogen-chemical properties methods to predict m6A sites based on support vector machine. Wang et al. [26] developed a new tool RFAthM6A to predict the N6-methyladenosine sites in Arabidopsis thaliana. Akbar et al. [27] developed the computational method iMethyl-STTNC, which used the STTNC to extract classification features and SVM as classifier. Liu et al. [28] constructed a m6A sites predictor using support vector machine algorithm. Qiang et al. [29] applied dinucleotide binary encoding and local position-specific dinucleotide frequency to extract features and used eXtreme Gradient Boosting (XGBoost) machine learning method for classification. Dao et al. [30] used physical–chemical property matrix, binary encoding, and nucleotide chemical property methods to extract RNA sequences information and SVM to construct m6A sites classifier.

Despite a number of m6A sites prediction methods based on ML are proposed, there is still some room for improvements. Firstly, multiple different feature extraction methods can be attempted to more comprehensively extract features of nucleotide sequences. Secondly, feature fusion inevitably brings redundancy and noise information. Choosing a suitable feature selection method can eliminate redundant features while retaining valid feature subset. Finally, a large amount of N6-methyladenosine data have been produced with the development of experimental techniques, so it is necessary to propose an effective prediction method based on a great deal of experimental data to improve the accuracy of m6A site prediction.

Inspired by the aforementioned descriptions, this paper put forward a novel prediction method DNN-m6A based on the deep neural network (DNN) to accurately predict m6A sites in different tissues of various species. Firstly, to efficiently convert nucleotide sequence character information into numerical vectors, we employ different feature extraction methods including binary encoding (BE), tri-nucleotide composition (TNC), enhanced nucleic acid composition (ENAC), *K*-spaced nucleotide pair frequencies (KSNPFs), nucleotide chemical property (NCP), pseudo dinucleotide composition (PseDNC), position-specific nucleotide propensity (PSNP), and position-specific dinucleotide propensity (PSDP) to extract multiple RNA sequence information and fuse the encoded feature vectors to obtain the initial feature vector sets of the benchmark datasets. Secondly, in order to preserve the effective features while deleting redundant and irrelevant features in the initial feature vector sets, we choose elastic net as the feature selection method to select the optimal feature subsets. Finally, the optimal feature subsets are input into DNN models whose hyper-parameters are optimized via TPE approach subsequently. After utilizing a variety of feature extraction methods, feature selection method and hyper-parameter optimization algorithm, the optimal parameters are used to construct the eleven tissues-specific classification models. In addition, the comprehensive comparison results on the training datasets and the independent datasets indicate that the prediction performance and generalization ability of DNN-m6A outperform the state-of-the-art method iRNA-m6A.

## 2. Materials and Methods 

### 2.1. Benchmark Datasets

Choosing the high-quality datasets is a key step in training an accurate and robust prediction model. In this paper, the RNA methylation modification datasets of three different genomes are downloaded from Dao’s work [30]. These benchmark datasets belong to different tissues of human (brain, liver, and kidney), mouse (brain, liver, heart, testis, and kidney), and rat (brain, liver, and kidney). The number of RNA positive samples and negative sample sequences in the datasets is equal, which gets rid of the influence of skewed datasets on the construction of robust models. The constructed datasets satisfy the following conditions: (1) The benchmark dataset is derived from the research of Zhang et al. [31]. To recognize the m6A sites in various tissues of different species, they developed m6A-REF-seq which is a precise and high-throughput antibody-independent m6A identification method based on the m6A-sensitive RNA endoribonuclease recognizing ACA motif [31]; (2) All the RNA fragment samples are 41-nt long with Adenine in the center site; (3) Using the CD-HIT program [32,33] to reduce sequence homology bias and remove sequences with sequence similarity more than 80%. In this paper, a sequence taking the m6A sites as the center site is referred as a positive sample. To objectively evaluate pros and cons of the built models, the datasets are divided into training datasets and independent datasets. Training dataset is used to select the optimal feature selection method and hyper-parameters of the model, and independent dataset is employed to examine the performance and generalization ability of the built model. The detailed information of the aforementioned positive and negative samples is given in Table 1. For the convenience of follow-up study, eleven datasets including human (brain, kidney, and liver), mouse (brain, heart, kidney, liver, and testis), and rat (brain, kidney, liver) are denoted by H_B, H_K, H_L, M_B, M_H, M_K, M_L, M_T, R_B, R_K, and R_L, respectively. 

### 2.2. Feature Extraction

It is a critical step to convert RNA sequence information into a numeric vector via feature extraction methods in the classification task, which directly influences the prediction performance of the model. In this study, we use nucleotide composition (NC) [34,35,36,37,38,39], *K*-spaced nucleotide pair frequencies (KSNPFs) [20,39,40], nucleotide chemical property (NCP) [37,40,41,42,43], binary encoding (BE) [20,30,36,37,44], pseudo dinucleotide composition (PseDNC) [34,37,39,45], and position-specific propensity (PSP) (including position-specific nucleotide propensity (PSNP) and position-specific dinucleotide propensity (PSDP)) [35,46,47,48] to extract the RNA sequence features.

#### 2.2.1. Binary Encoding (BE)

Binary encoding is a common encoding mean which can exactly depict the nucleotides at each position in the sample sequence. For each nucleotide in the RNA sequence will be encoded as a 4-dimensional binary vector according to the following rules: ‘;adenine (A)’-> 1000, ‘cytosine (C)’-> 0100, ‘guanine (G)’-> 0010, and ‘uracil (U)’-> 0001 (e.g., the RNA sequence ‘GGAUUCGA’ is represented as ${\left[00100010......1000\right]}^{T}$). Thus, a 41-nt long RNA sequence sample will be converted into a 164 (41 × 4) dimensional feature vector.

#### 2.2.2. Nucleotide Composition (NC)

Nucleotide composition (NC) (i.e., *K*-mer nucleotide frequency), is a classic coding method for expressing the features of a nucleotide sequence, which is used to calculate the frequency of occurrence for each *K*-mer nucleotide in the sample sequence and will generate a $4^{K}$-dimensional feature vector. In this study, tri-nucleotide composition (TNC) and enhanced nucleic acid composition (ENAC) are employed to encode the sample sequence. TNC and ENAC are corresponding to 3-mer nucleotide frequency and a variation method of 1-mer nucleotide frequency (i.e., NAC), respectively. ENAC figures NAC based on a fixed-length sequence window that slides from the 5′; to 3′ terminus of each RNA sequence in succession, in which the length of the window is set to 5. Moreover, the following formula is used to calculate *K*-mer nucleotide frequency:(1)$$f(n_{1}n_{2}\ldots n_{K})=\frac{N(n_{1}n_{2}\ldots n_{K})}{\left(L-K+1\right)},(n_{K}\in (A,C,G,U))$$
where $n_{1}n_{2}\ldots n_{K}$ indicates a *K*-mer nucleotide component, and $N(n_{1}n_{2}\ldots n_{K})$ is the number of occurrences of $n_{1}n_{2}\ldots n_{i}\ldots n_{K}$ in an RNA sequence. By the TNC and ENAC feature extraction methods, an RNA sequence sample of 41-nt long will be encoded as a 64-dimensional and a 148-dimensional feature vectors, respectively.

#### 2.2.3. K-Spaced Nucleotide Pair Frequencies (KSNPFs)

The KSNPFs feature encoding is used to calculate the frequency of nucleotide pairs separated by *K* arbitrary nucleotides. For illustrative purposes, $N_{1}x\{K{\}N}_{2}\;(N_{1}{,N}_{2}\;\mathrm{and}\;x\in \{A,C,G,U\})$ is used to denote *K*-spaced nucleotide pairs. For instance, AxxC is a two-spaced nucleotide pair in which two arbitrary nucleotides are between the nucleotides A and C. The feature vector of KSNPFs is defined as follows:(2)$${\left(\frac{N(\mathrm{Ax}\{K\}A)}{L-K-1},\frac{N(\mathrm{Ax}\{K\}C)}{L-K-1},\ldots ,\frac{N(\mathrm{Ux}\{K\}U)}{L-K-1}\right)}_{16}$$
where $N(N_{1}x\{K{\}N}_{2})$ represents the number of $N_{1}x\{K{\}N}_{2}$ in an RNA sequence. For example, when 5 is selected as the optimal value of the $K_{max}$ (i.e., *K*=0, 1, 2, 3, 4, 5), then the 96 (4×4×(5+1)) dimensional feature vector is obtained. Accordingly, an RNA sequence of 41-nt can generate a vector of ($4\times 4\times (K_{max}+1)$) dimensional when the value of the parameter $K_{max}$ is determined.

#### 2.2.4. Position-Specific Nucleotide Propensity (PSNP) and Position-Specific Dinucleotide Propensity (PSDP)

Position-specific propensity (PSP) is an encoding method used to calculate the frequency of nucleotides at certain positions and extract statistical information from sequences. For an RNA sequence $R=N_{1}N_{2}N_{3}\ldots N_{L}$, its details of the nucleotide position specificity can be formulated by the following 4×L matrix:(3)$$Z_{PSNP}=\left[\begin{array}{cccc} z_{1,1} & z_{1,2} & \cdots & z_{1,L}\\ z_{2,1} & z_{2,2} & \cdots & z_{2,L}\\ z_{3,1} & z_{3,2} & \cdots & z_{3,L}\\ z_{4,1} & z_{4,2} & \cdots & z_{4,L} \end{array}\right]$$
where
(4)$$z_{i,j}=z_{i,j}^{+}-z_{i,j}^{-},(i=1,\ldots ,4;j=1,\ldots ,L)$$
$z_{i,j}^{+}$ and $z_{i,j}^{-}$ denote the frequency of occurrence of the *i*-th nucleotide at the *j*-th position in the positive $\left(S^{+}\right)$ and the negative $\left(S^{-}\right)$ dataset, respectively. Hence, an RNA sequence of *L*-nt can be expressed as
(5)$$S_{PSNP}={\left[f_{1}\;f_{2}\;\cdots \;f_{j}\;\cdots \;f_{L}\right]}^{T}$$
where *T* is the operator of transpose, and $f_{j}$ is defined as
(6)$$f_{j}=\{\begin{array}{cc} z_{1,j}, & when\;N_{j}=A\\ z_{2,j}, & when\;N_{j}=C\\ z_{3,j}, & when\;N_{j}=G\\ z_{4,j}, & when\;N_{j}=U \end{array},\;(j=1,2,\ldots ,L)$$

Similarly, following the principle used to generate the $Z_{PSNP}$ matrix, we can obtain the 16×(L−1) position-specific dinucleotide propensity (PSDP) matrix:(7)$$Z_{PSDP}=\left[\begin{array}{cccc} z_{1,1} & z_{1,2} & \cdots & z_{1,L-1}\\ z_{2,1} & z_{2,2} & \cdots & z_{2,L-1}\\ \vdots & \vdots & \ddots & \vdots \\ z_{16,1} & z_{16,2} & \cdots & z_{16,L-1} \end{array}\right]$$

The corresponding feature vector can be expressed as
(8)$$S_{PSDP}=[f_{1}\;f_{2}\;\cdots \;f_{j}\;\cdots \;f_{L-1}]$$
where each element $f_{j}$ is obtained from the $Z_{PSDP}$ matrix in (8), which is defined as follows:(9)$$f_{j}=\{\begin{array}{cc} z_{1,j}, & when\;N_{j}N_{j+1}=AA\\ z_{2,j}, & when\;N_{j}N_{j+1}=AC\\ & \vdots \\ z_{16,j}, & when\;N_{j}N_{j+1}=UU \end{array},\;(j=1,2,\ldots ,L-1)$$

Through the PSNP and PSDP feature extraction methods, an RNA sequence sample will be encoded by a 41-dimensional and a 40-dimensional feature vectors, respectively.

#### 2.2.5. Nucleotide Chemical Property (NCP)

RNA consists of four different kinds of nucleic acids: adenine (A), guanine (G), cytosine (C) and uracil (U). They can be categorized into three different groups in terms of different chemical properties (Table 2): (1) from the angle of ring structures, adenine and guanine are purines containing two rings, while cytosine and uracil only one; (2) from the perspective of functional group, adenine and cytosine pertain to amino group, whereas guanine and uracil to keto group; (3) from the angle of hydrogen bond, strong hydrogen bonds are possessed by guanine and cytosine, but adenine and uracil have weak one.

According to the foregoing three partition methods, A, C, G, and U can be expressed in the coordinates (1, 1, 1), (0, 1, 0), (1, 0, 0), and (0, 0, 1), respectively. Therefore, an RNA sequence of 41-nt long will be encoded by a 123 (41 × 3) dimensional vector.

#### 2.2.6. Pseudo Dinucleotide Composition (PseDNC)

Pseudo dinucleotide composition (PseDNC) is a feature extraction method which can merge the local sequence order information and global sequence order information into the feature vector of the RNA sequences. The feature vector *D* generated by PseDNC can be used to define a given RNA sequence:(10)$$D={\left[d_{1},d_{2},\ldots ,d_{16},d_{16+1},\ldots ,d_{16+\lambda }\right]}^{T}$$
where
(11)$$d_{k}=\{\begin{array}{c} \frac{f_{k}}{\sum_{i=1}^{16}f_{i}+w\sum_{j=1}^{\lambda }\theta_{j}},(1\leq k\leq 16)\\ \frac{w\theta_{k-16}}{\sum_{i=1}^{16}f_{i}+w\sum_{j=1}^{\lambda }\theta_{j}},(17\leq k\leq 16+\lambda ) \end{array}$$

In Equation (11), $f_{k}$ represents the normalized occurrence frequency of non-overlapping dinucleotides at the *k*-th position in the RNA sequence. *λ* is an integer indicating the highest counted tie (or rank) of the correlation along the RNA sequence, and *w* denotes the weight factor ranged from 0 to 1. $\theta_{j}$ represents the *j*-tier correlation factor calculated by (12) and (13).
(12)$$\theta_{j}=\frac{\sum_{i=1}^{L-j-1}\theta (R_{i}R_{i+1},R_{i+j}R_{i+j+1})}{L-j-1},(1\leq j\leq \lambda ;\lambda <L)$$
(13)$$\theta (R_{i}R_{i+1},R_{i+j}R_{i+j+1})=\frac{\sum_{u=1}^{\mu }\left[P_{u}(R_{i}R_{i+1})-P_{u}(R_{i+j}R_{i+j+1}\right){]}^{2}}{\mu }$$
where *µ* suggests the number of RNA physicochemical indices. Six indices including “Rise”, “Roll”, “Shift”, “Slide”, “Tilt”, and “Twist” are set as the indices for RNA sequences. $P_{u}(R_{i}R_{i+1})$ is the *u*-th (u=1,2,…,μ) physicochemical index’s numeral value of dinucleotide $R_{i}R_{i+1}$ at the position *i*, and $P_{u}(R_{i+1}R_{i+j+1})$ illustrates the corresponding value of the dinucleotide $R_{i+1}R_{i+j+1}$ at position i+j. According to PseDNC, (16+*λ*) dimensional feature vector can be obtained for each RNA sequence.

### 2.3. Feature Selection Method

Elastic net (EN) [49] is a method that can perform feature selection based on regularized term. Its regularization term is a mixture of ridge regression’s regularization term and lasso regression’s regularization term and the mixing percentage is controlled by the parameter *β*. Namely, when *β* = 0, EN is equivalent to ridge regression [50], while *β* = 1 is up to lasso regression [51]. The objective function of the EN can be defined as follows:(14)$$\underset{w}{\min}\frac{1}{2\times n}||y-Xw|{|}_{2}^{2}-\alpha \times \beta ||w|{|}_{1}+\frac{1}{2}\alpha \times (1-\beta )||w|{|}_{2}^{2}$$
where *X* is the sample matrix, *y* is the category label, *α* and *β* are non-negative penalty parameters, *w* represents the regression coefficient, and *n* indicates the number of samples.

### 2.4. Deep Neural Network

A neural network is a mathematical model consisting of an input layer, multiple intermediate hidden layers and an output layer. A deep neural network (DNN) is a neural network with two or more hidden layers. In addition to the output layer, each layer in the DNN is fully connected to the next layer. The given feature matrix is first received in the input layer, and then non-linearly converted across multiple hidden layers. The following mathematical expression is used to denote the input data of the layer *l*:(15)$$a^{l}=\delta (w^{l}a^{l-1}+b^{l}),(l=1,2,\ldots ,N)$$
where $w^{l}$ and $b^{l}$ are the connection weight matrix and the bias of the layer, respectively. δ represents the non-linear activation function of the *l*-th layer. The cross-entropy loss function is used to optimize the models, which is defined as
(16)$$L=-\frac{1}{N}\sum_{i=1}^{N}\left(y_{i}\log{\hat{y}}_{i}+(1-y_{i})\log(1-{\hat{y}}_{i})\right)$$
where *N* is the number of samples, $y_{i}$ denotes the true label, and ${\hat{y}}_{i}$ represents the predictive label.

In the last layer, for classification, the sigmoid function $\sigma (x)=1/1+e^{-x}$ is employed as the nonlinear transformation to map the output to the interval [0,1]. Moreover, a dropout layer is employed following each hidden layer to avoid over-fitting, and the hyper-parameters of models are optimized accordingly in Section 3.4.

### 2.5. Hyper-Parameter Optimization

The TPE approach is a Bayesian optimization algorithm under the framework of SMBO, which achieves better results in several difficult learning problems [52]. TPE is based on p(λ|c) and p(c) to model p(c|λ) indirectly. p(λ|c) is defined as follow:(17)$$p(c|\lambda )=\{\begin{array}{c} l(x),c<c^{*}\\ g(x),c\geq c^{*} \end{array}$$
where *c* is the loss under the hyper-parameter setting *λ*, *c** denotes a predefined threshold value, which is typically set as a *γ*-quantile of the best-observed *c* [53]. Let $\left\{\lambda^{\left(1\right)},\ldots ,\lambda^{\left(k\right)}\right\}$ be the different observations, l(λ) is the density estimate formed by the observations $\left\{\lambda^{\left(i\right)}\right\}$ so that corresponding loss value $c(\lambda^{\left(i\right)})$ is lower than *c**, whereas g(λ) is produced by the remaining observations [54]. In order to determine the settings of the local optimal hyper-parameter, expected improvement (EI) is chosen as acquisition function, and Bergstra et al. [54] have demonstrated that EI in (18) is proportional to the following expression:(18)$$EI(\lambda )\propto {\left(\gamma +\frac{g(\lambda )}{l(\lambda )}(1-\gamma )\right)}^{-1}$$

This expression suggests that hyper-parameter *λ* should have a high probability under l(λ) and a low probability under g(λ) for the sake of maximizing EI [53]. In addition, the l(λ) and g(λ) of tree-structured form make it easier for TPE to gain candidate hyper-parameters than other SMBO approaches. Consequently, the Bayesian hyper-parameter optimization approach TPE is employed to tune the hyper-parameters of the DNN models in this study.

### 2.6. Performance Evolution

In statistical prediction, *K*-fold cross-validation, jackknife validation test, and independent dataset test are normally used to evaluate models. In this paper, we use 5-fold cross-validation to assess the effectiveness of the model. For the sake of proving the robustness of model ulteriorly, the independent dataset is used to test it after its establishment. To measure the performance of models more intuitively, sensitivity (Sn), specificity (Sp), accuracy (ACC) and Matthew’s correlation coefficient (MCC) are used as evaluation indicators, which are defined as
(19)$$Sn=\frac{TP}{TP+FN},(0\leq Sn\leq 1)$$
(20)$$Sp=\frac{TN}{TN+FP},(0\leq Sp\leq 1)$$
(21)$$ACC=\frac{TP+TN}{TP+TN+FP+FN},(0\leq ACC\leq 1)$$
(22)$$MCC=\frac{TP\times TN-FP\times FN}{\sqrt{\left(TP+FN)\times (TN+FN)\times (TP+FP)+(TN+FP\right)}},(-1\leq MCC\leq 1)$$
where *TP*, *TN*, *FP,* and *FN* respectively stand for the number of true positives, true negatives, false positives, and false negatives. In addition, the area under the curve (AUC) [55] is also an important index to evaluate the predictive performance of models. AUC represents the area under the receiver operator characteristic (ROC) curve. The larger the AUC, the better the performance of the model, and AUC = 1 means a perfect model.

### 2.7. Description of the DNN-m6A Process

In this study, we propose a novel method called DNN-m6A for identifying m6A sites, whose flowchart is shown in Figure 1. All the experiments are executed on Windows operating system with 32.0 GB of RAM, and implemented by Python 3.7 programming. The Deep Neural Network model is carried out through Keras (version 2.4.3) which is provided by Tensorflow (version 2.3.1). The specific steps of the DNN-m6A method proposed in this study are described as:

(1) Obtain the datasets. The eleven benchmark datasets of different tissues from three species are obtained, and then the nucleotide sequences and corresponding class labels of the positive and negative samples are input into models.

(2) Feature extraction. Firstly, the DNN model with ReLU activation function, Adam optimizer and two hidden layers is used to determine the optimal parameters in the KSNPFs and PseDNC feature extraction methods through 5-fold cross-validation test. Subsequently, the encoded feature vectors extracted by BE, KSNPFs, ENAC, NCP, PseDNC, TNC, PSNP, and PSDP are fused to obtain the initial feature vector sets.

(3) Feature selection. For the initial feature vector sets, the EN based on L1 and L2 regularization is applied to eliminate redundant and irrelevant information, and retain the optimal feature vectors aiding to classification.

(4) Hyper-parameter optimization. The optimal feature subsets obtained via Step 2 and Step 3 are used as the input features of the models, and then the TPE is used to optimize the hyper-parameters of the DNN models.

(5) Model evaluation. Compute the AUC, ACC, Sn, Sp, and MCC via 5-fold cross-validation on the training datasets to assess the predictive performance of the classification models.

(6) Using the models constructed in Step 1–Step 5, and the independent datasets are used to test the effectiveness and robustness of the models.

## 3. Results and Discussion

### 3.1. Parameter Selection of Feature Extraction

The feature extraction methods need to select the optimal parameters, which have a vital effect on the construction of prediction models. In this study, the parameters *λ* and *w* of PseDNC, and *K_max_* of KSNPFs can generate influence to the performance of classification models. With the increasing of the values of *λ* and *K_max_*, the more information will be sufficiently extracted, while this will also produce redundant features. Thus, when selecting the optimal parameter *K_max_* in KSNPFs, the value of *K_max_* is set from 1 to 5 with an interval of 1. Considering the nucleotide sequence length in the datasets is 41, we search for the best values of the two parameters in the range of w∈[0.1,0.9] and λ∈[10,30] with steps of 0.2 and 10, respectively. The feature matrices with different parameters are served as the input features of the DNN model with ReLU activation function, Adam optimizer and two hidden layers. The prediction accuracy values of the eleven different tissue datasets under different parameters are shown in Appendix A, in which the prediction accuracy values are calculated by DNN model in 5-fold cross-validation test. The influence of different *λ* and *w* of PseDNC, and *K_max_* of KSNPFs on ACC is shown in Figure 2 and Figure 3, respectively.

From Figure 2, we can intuitively see that for eleven datasets of various tissues from different species, the corresponding *λ* and *w* values are inconsistent when the ACC reaches the highest. For example, for the datasets H_L and R_K, when *λ* = 30 and *w* = 0.3, the accuracy reaches the maximum value of 77.13% and 80.79%, respectively; for the datasets M_K and R_B, when *λ* = 30 and *w* = 0.5, the accuracy reaches the highest value of 78.99% and 73.77%, respectively; and for the datasets M_L and M_T, when *λ* = 20 and *w* = 0.9, the ACC reaches the maximum of 68.03% and 71.87%, respectively (Appendix A). In addition, as shown in Figure 3, we can see ACC values of the eleven datasets are different with the change of parameter *K_max_* values. When using KSNPFs to extract features, the ACC values of the eleven datasets change with the change of the *K_max_* value. For example, for the datasets H_B, H_K, M_L, R_K, and R_L, when the *K_max_* is 3, the ACC reaches the highest value of 70.14%, 77.54%, 70.23%, 81.56%, and 80.76%, respectively; meanwhile, the accuracy values of the datasets H_L, M_B, M_K, and R_B are the highest when the parameter *K_max_* value is 5 (Appendix A). In order to construct the optimal eleven tissue-specific models, two methods (i.e., KSNPFs and PseDNC) whose optimal parameters are determined in each dataset are used to extract the feature matrices.

### 3.2. The Performance of Feature Extraction Methods

After determining the optimal parameters of PseDNC and KSNPFs, BE, KSNPFs, ENAC, NCP, PseDNC, TNC, PSNP, and PSDP are fused to gain a more comprehensive information. In order to measure the differences between BE, KSNPFs, ENAC, NCP, PseDNC, TNC, PSNP, and PSDP, the eight individual feature sets and fusion features are fed into the DNN model. Figure 4 shows the ACC values for different feature extraction methods on the eleven tissues’ training datasets, which are obtained via fivefold cross-validation. In Figure 4, “All” denotes the result of multi-information fusion. 

It can be seen from Figure 4 that various feature extraction methods enable to different datasets obtain different prediction accuracy. For the datasets M_L, M_T, and R_K, when using the TNC feature extraction method, the accuracy reaches the maximum value of 70.99%, 73.94%, and 81.59%, respectively; for the datasets M_H, M_K, and R_L, the accuracy is 72.63%, 79.86%, and 80.76% which reach the maximum respectively when using the KSNPFs feature extraction method; and for the datasets H_L and R_B, using the PSDP feature extraction method make the ACC reaches the maximum of 78.93% and 75.38%, respectively (Appendix A). For the datasets H_B, H_K, and M_B, the prediction accuracy of fusion features is better than that of individual feature set. While for other datasets, the accuracy after fusion of multiple information is a little worse than the accuracy of prediction using some single set of features. It indicates that fusion features can improve the prediction accuracy of the model to a certain extent, but with the increasing of the dimension of the feature vector, multi-information fusion will unavoidably bring redundant information.

### 3.3. The Performance of Feature Selection Methods

Fusing BE, KSNPFs, ENAC, NCP, PseDNC, TNC, PSNP, and PSDP can construct raw feature spaces, which are used to build eleven DNN classification models. It should be noted that multi-information fusion can obtain more comprehensive information, but it also inevitably brings redundant feature information, which affects the prediction accuracy of the models and reduces the calculation speed. Consequently, it is necessary to utilize the feature selection method that can eliminate redundancy and noise information and obtain the optimal feature subset. When using the EN method for feature selection, disparate *α* values make the model produce different prediction results. In order to select the optimal feature subsets for each dataset, we set the value of the parameter *α* to 0.01, 0.02, 0.03, 0.04, 0.05, 0.06, 0.07, 0.08, 0.09, and 0.1 in turn. The ACC values of the datasets M_T, M_B, and M_H reach the maximum when the parameter *α* is 0.03, 0.05, and 0.07, respectively. The ACC values of the datasets M_K, R_B and R_L reach the maximum when the parameter α is 0.1, which are 81.51%, 78.19% and 82.29%, respectively. Meanwhile, the ACC values of the datasets H_B, H_K, H_L, M_L and R_K reach the maximum when the parameter *α* value is 0.09. By setting different *α* values in the EN method, we can obtain the optimal feature subsets for eleven different tissues datasets, which can maximize the accuracy of the models. In addition, we also use Locally linear embedding (LLE) [56], minimum redundancy maximum relevance (mRMR) [57], spectral embedding (SE) [58], and singular value decomposition (SVD) [59] to eliminate redundant information. In order to better compare with EN, the feature subsets corresponding to the feature selection methods of LLE, mRMR, SE, SVD on the eleven tissues’ training datasets are set to the same feature dimensions as the EN method. The comparison of the prediction results and dimensions of five feature selection methods for the eleven datasets are shown in Appendix A. In Appendix A, “Initial” represents the dimension of initial feature vector sets without doing feature reduction, “Optimal” denotes the dimension of the optimal feature subsets. The graphical illustration of the experimental results of different feature selection method is shown in Figure 5.

It can be seen from Figure 5 that different feature selection methods have different dimensional reduction effects on the initial feature spaces of the eleven datasets. EN has a better effect compared with the other four feature selection methods, whose ACC values corresponding to the datasets H_B, H_K, H_L, M_B, M_H, M_K, M_L, M_T, R_B, R_K, and R_L reach 73.44%, 79.84%, 80.77%, 78.90%, 75.65%, 81.51%, 73.03%, 76.16%, 78.19%, 83.21%, and 82.29%, respectively. They are 6.40%, 4.14%, 3.68%, 4.30%, 6.09%, 3.66%, 7.24%, 6.42%, 6.61%, 4.62%, and 3.94% higher than the values corresponding to LLE, respectively. Meanwhile, the ACC values of mRMR are 1.92%, 0.19%, 0.55%, 0.45%, 2.57%, 1.15%, 2.49%, 1.57%, 1.30%, 0.85%, and 0.73% lower than EN, respectively (Appendix A). Furthermore, we can clearly see that the models trained on the optimal feature subsets obtained by EN method have improved performance compared to models trained on the initial feature sets. However, compared with the models trained on the complete feature sets, the ACC and AUC values of the models trained on the feature subsets obtained by LLE method are reduced, and some similar situations also occur when using the SE and SVD methods.

To intuitively compare the prediction performance of the five feature selection methods on different datasets, the ROC curves for different feature selection methods are shown in Figure 6. For the datasets H_B, H_K, H_L, M_B, M_H, M_K, M_L, M_T, R_B, R_K, and R_L, the AUC values corresponding to the EN feature selection method are 0.8131, 0.8826, 0.8859, 0.8758, 0.8375, 0.8944, 0.8114, 0.8429, 0.8672, 0.9087, and 0.8962, respectively. The AUC values corresponding to the datasets H_B, H_K, H_L, M_B, and M_H are 7.35%, 4.52%, 4.24%, 5.18%, and 7.83% higher than the values corresponding to LLE, respectively. The AUC values corresponding to the datasets M_K, M_L, M_T, R_B, R_K, and R_L are 1.00%, 2.94%, 1.14%, 1.56%, 0.75%, and 0.60% higher than the values corresponding to mRMR, respectively. Given the above, the performance of the m6A sites prediction models constructed by the EN methods achieve excellent results as compared with other feature selection methods. Therefore, we choose EN to eliminate redundant information that has little correlation with m6A sites, and retain feature subsets contributed to classification, thereby affording effective feature fusion information for DNN models.

### 3.4. The Performance of Hyper-Parameter Optimization

According to the analysis in the Section 3.3, EN is used as the feature selection method to construct the optimal feature subsets, which are acted as the trainset for training eleven tissues-specific models. To further improve the performance of the DNN models, the TPE approach, which is a Bayesian optimization algorithm, is adopted to optimize some critical hyper-parameters in eleven classification models. The optimization ranges and results of the hyper-parameters are shown in Table 3 and Table 4, respectively. The accuracy value between experimental values and predictive values of fivefold cross-validation is defined as the fitness function evaluations of hyper-parameters optimization of DNN models. The prediction performance metrics of the final models acquired after optimizing the hyper-parameters via TPE approach are exhibited in Table 5.

As can be seen from Table 5, for eleven datasets, compared with the models without carrying out hyper-parameter optimization, the prediction performance of the new models has been significantly improved. The maximum ACC values of the datasets H_B, H_K, H_L, M_B, M_H, M_K, M_L, M_T, R_B, R_K, and R_L obtained after hyper-parameter optimization are 73.78%, 80.48%, 81.30%, 79.36%, 76.17%, 81.96%, 73.58%, 76.62%, 78.27%, 83.38%, and 82.63%, respectively. The AUC values corresponding to the datasets are 0.34%, 0.15%, 0.46%, 0.20%, 0.64%, 0.09%, 0.25%, 0.64%, 0.06%, 0.17%, and 0.29% higher than the values corresponding to models without hyper-parameter optimization, respectively. Hence, careful tuning of these hyper-parameters can further improve the performance of the models. Ultimately, the eleven final prediction models are constructed after hyper-parameter optimization by TPE.

### 3.5. Comparison of DNN-m6A with Other State-of-the-Art Methods 

For the sake of validating the effectiveness of the DNN-m6A method in identifying m6A sites, we compare our proposed method with the state-of-the-art predictor iRNA-m6A [30]. In iRNA-m6A, physical–chemical property matrix, mono-nucleotide binary encoding, and nucleotide chemical property were used to extract features, and the classification models were constructed by SVM in fivefold cross-validation test. The prediction results comparison of DNN-m6A and iRNA-m6A on the training datasets are shown in Figure 7 and the detailed comparison results are shown in Appendix A.

From Figure 7, we can intuitively see that the prediction results of DNN-m6A are higher than that of iRNA-m6A, for eleven tissues’ training datasets (i.e., H_B, H_K, H_L, M_B, M_H, M_K, M_L, M_T, R_B, R_K, and R_L). DNN-m6A’s ACC values for eleven datasets are 2.52%, 1.49%, 1.17%, 0.61%, 3.41%, 1.98%, 2.99%, 2.22%, 2.31%, 1.60%, and 1.73% higher than that of iRNA-m6A, respectively (Appendix A). Additionally, the AUC values of DNN-m6A reach 81.65%, 88.41%, 89.05%, 87.78%, 84.39%, 89.53%, 81.39%, 84.93%, 86.78%, 91.04%, and 89.91%, respectively, which is 4.09%, 2.07%, 1.67%, 0.77%, 4.91%, 2.27%, 3.96%, 3.37%, 3.96%, 2.27%, and 2.25% higher than that of iRNA-m6A (Appendix A). To prove the generalization ability and robustness of DNN-m6A ulteriorly, the independent datasets are applied to test our DNN-m6A method. Additionally, the test results are compared with the existing method subsequently. The feature extraction parameters, feature selection method and classifier parameters of the independent datasets are strictly in keeping with the training datasets. The results of the comparison with iRNA-m6A on the independent datasets are shown in Appendix A, and the graphical illustration of the experimental results is shown in the Figure 8.

It can be observed that the proposed DNN-m6A method achieves better performance than iRNA-m6A method from Figure 8. More specifically, for the eleven independent datasets, the ACC of DNN-m6A reaches 73.27%, 79.89%, 80.96%, 78.59%, 75.11%, 80.87%, 72.95%, 77.12%, 77.99%, 83.04%, and 81.64%, respectively, which is 2.17%, 2.13%, 1.95%, 0.33%, 3.81%, 1.56%, 4.16%, 3.58%, 2.85%, 1.62%, and 1.79% higher than the iRNA-m6A (Appendix A). Meanwhile, the AUC values of DNN-m6A are 3.02%, 2.15%, 1.73%, 1.47%, 4.60%, 1.91%, 4.59%, 3.53%, 3.59%, 1.41%, and 1.95% higher than the values corresponding to the iRNA-m6A, respectively (Appendix A). These results indeed show the superiority of the DNN-m6A method is over the iRNA-m6A method. We further evaluated the efficacy of the proposed method using another benchmark dataset, i.e., S51. The dataset S51 was downloaded from M6AMRFS [29]. The comparison results of DNN-m6A with pRNAm-PC [28], M6AMRFS, and iN6-Methyl (five-step) [60] (using the same dataset and 10-fold cross-validation) are shown in Appendix A and Figure 9. As can be seen from Figure 9, ACC, Sn, Sp and MCC values of DNN-m6A exceed those of methods pRNAm-PC, M6AMRFS, and iN6-Methyl (5-step). Take ACC as an example, 8.76%, 4.25%, and 3.12% improvements were observed compared with pRNAm-PC, M6AMRFS, and iN6-Methyl (5-step) (Appendix A). In conclusion, the results indicate that the DNN-m6A method can remarkably improve the prediction accuracy of m6A sites and achieve excellent generalization ability.

## 4. Conclusions

Since m6A plays an important role in many biological processes, the accurate identification of m6A sites is essential for the basic research of RNA methylation modification. In this study, we put forward a novel method DNN-m6A for the detection of m6A sites in different tissues with high accuracy. For eleven benchmark datasets of different tissues from three species, we first employ feature extraction methods of BE, KSNPFs, ENAC, NCP, PseDNC, TNC, PSNP, and PSDP to extract features of RNA sequences and fuse eight groups of features to gain the original feature space. Secondly, elastic net is used to eliminate redundant and noise information from extracted vectors while keeping the effective features related to model classification. Finally, based on the optimal feature subset, the TPE approach is used to optimize the hyper-parameters of DNN models. We use the proposed DNN-m6A to construct the best models for the cross-species/tissues datasets through fivefold cross-validation so that it can be well used for the m6A site detection. ACC, MCC, Sn, Sp, and AUC are used to evaluate the performance of the models. Corresponding prediction accuracy of the eleven tissues’ training datasets (i.e., H_B, H_K, H_L, M_B, M_H, M_K, M_L, M_T, R_B, R_K, and R_L) are 73.78%, 80.48%, 81.30%, 79.36%, 76.17%, 81.96%, 73.58%, 76.62%, 78.27%, 83.38%, and 82.63%, respectively, which is 2.52%, 1.49%, 1.17%, 0.61%, 3.41%, 1.98%, 2.99%, 2.22%, 2.31%, 1.60%, and 1.73% better than the state-of-the-art method. Moreover, we introduce the independent datasets to further prove the superiority of this method. Comprehensive comparison results show that the method we propose has stronger competitiveness in the identification of m6A sites. The source code of the propose method is freely available at http://github.com/GD818/DNN-m6A (accessed on 27 February 2021).

## Figures and Tables

**Figure 1 genes-12-00354-f001:**
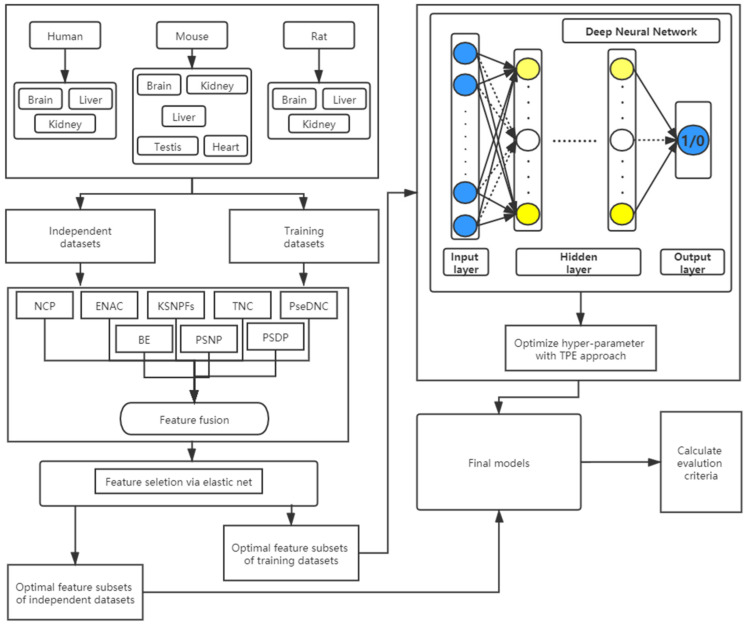
The flowchart of the DNN-m6A method.

**Figure 2 genes-12-00354-f002:**
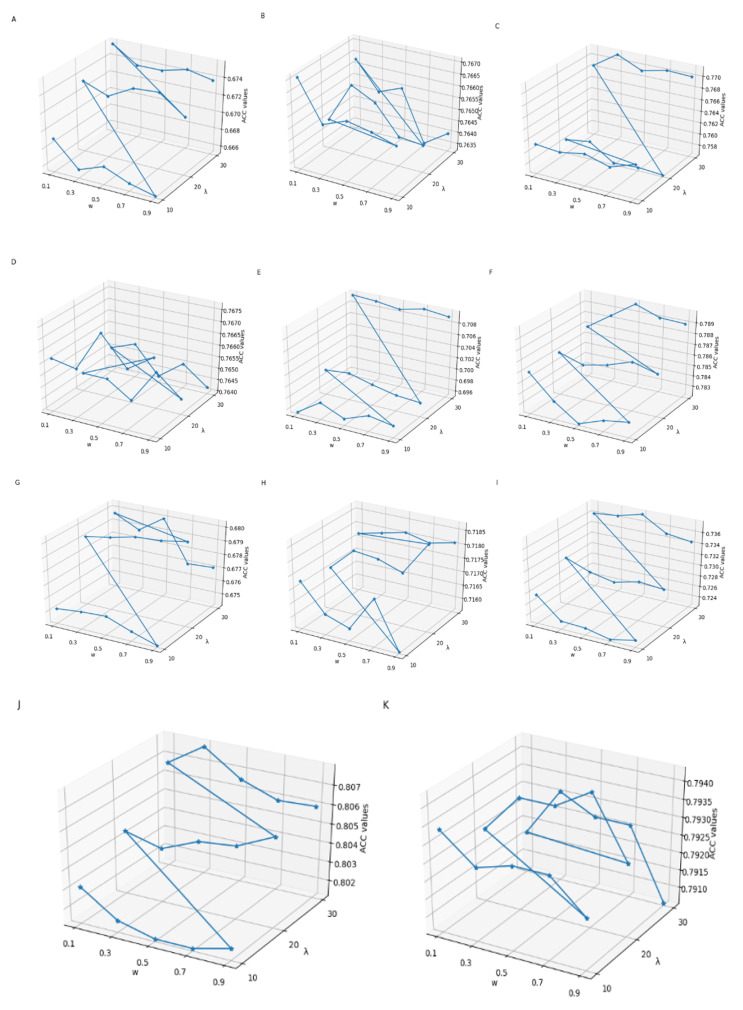
The effect of choosing different *λ* and *w* values in PseDNC on eleven datasets (**A**) H_B, (**B**) H_K, (**C**) H_L, (**D**) M_B, (**E**) M_H, (**F**) M_K, (**G**) M_L, (**H**) M_T, (**I**) R_B, (**J**) R_K, and (**K**) R_L.

**Figure 3 genes-12-00354-f003:**
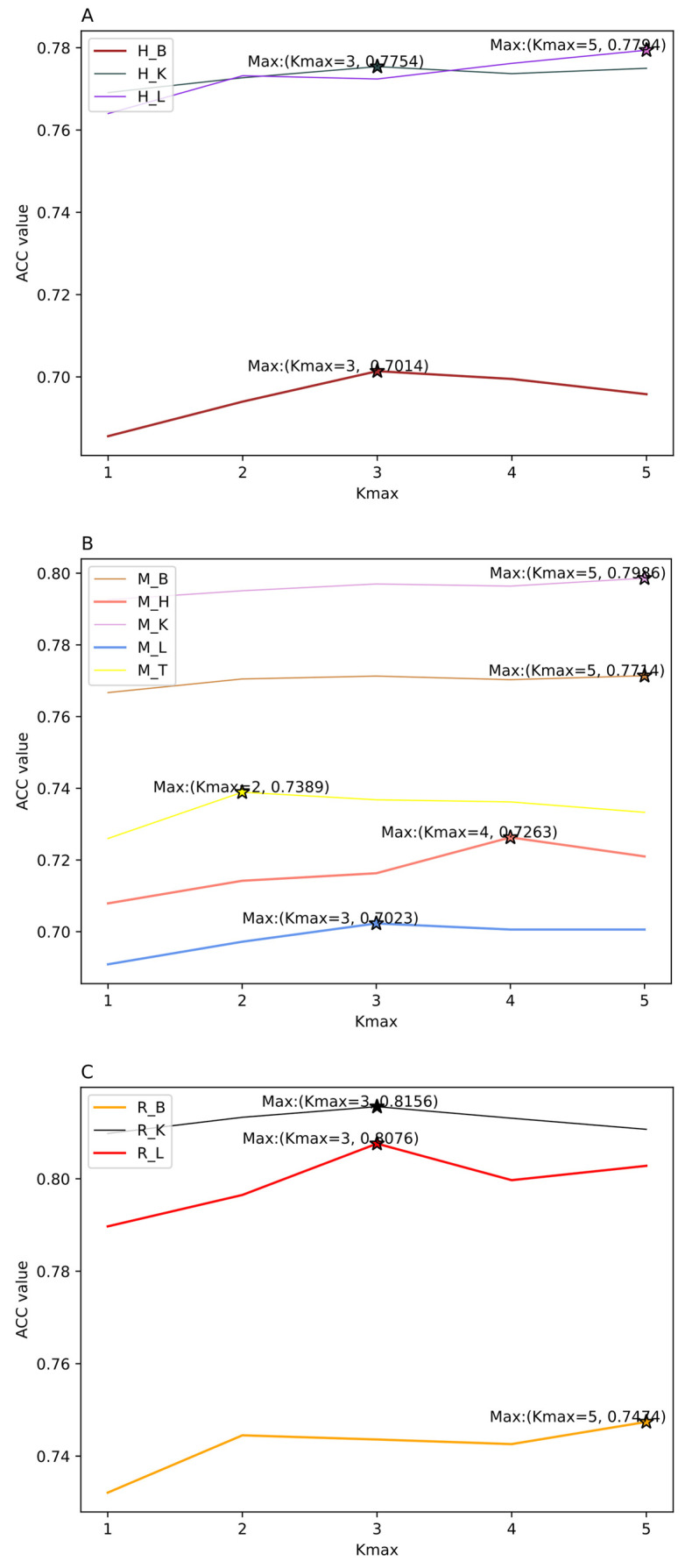
The effect of choosing different *K_max_* values in K-spaced Nucleotide Pair Frequencies (KSNPFs) on eleven datasets. Figure (**A**) shows the accuracy (ACC) values for different *K_max_* values on H_B, H_K, and H_L datasets. Figure (**B**) shows the ACC values for different *K_max_* values on M_B, M_H, M_K, M_L, and M_T datasets. Figure (**C**) shows the ACC values for different *K_max_* values on R_B, R_K, and R_L datasets.

**Figure 4 genes-12-00354-f004:**
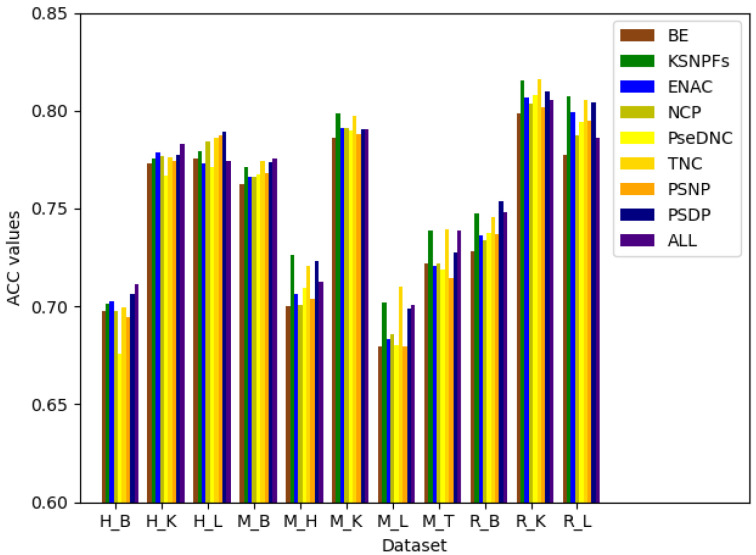
ACC values corresponding to different feature extraction methods.

**Figure 5 genes-12-00354-f005:**
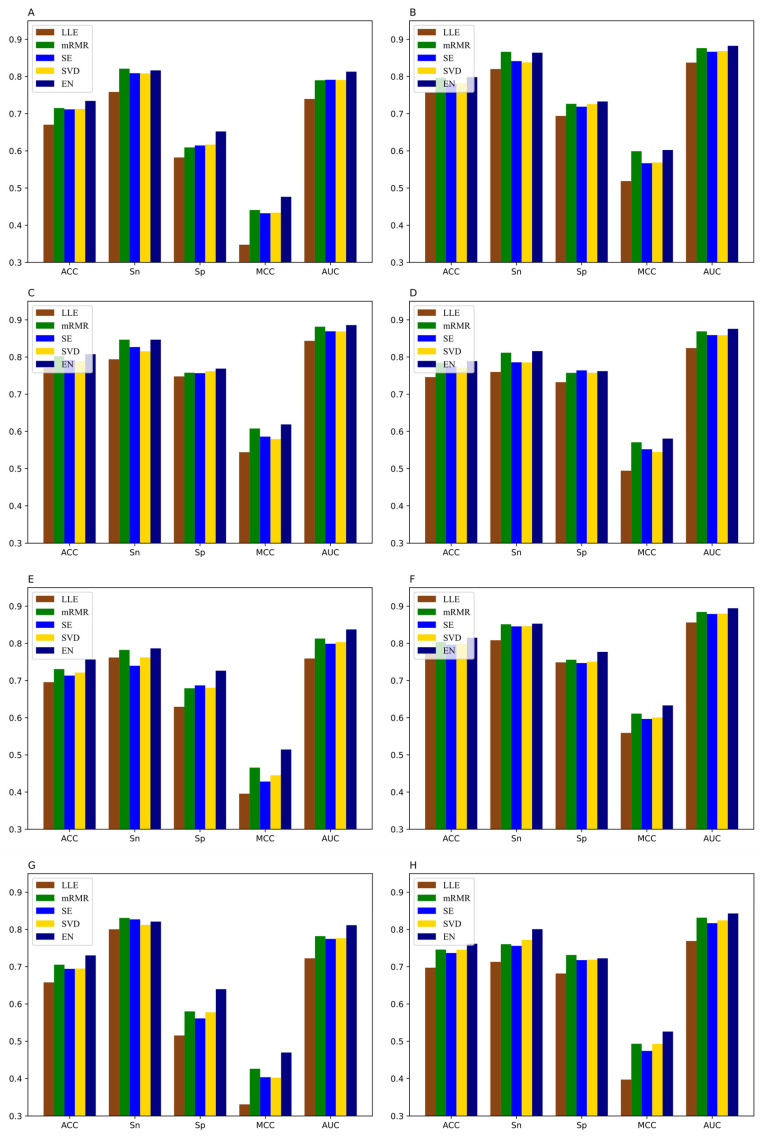
Performance of different feature selection methods on eleven datasets (**A**) H_B, (**B**) H_K, (**C**) H_L, (**D**) M_B, (**E**) M_H, (**F**) M_K, (**G**) M_L, (**H**) M_T, (**I**) R_B, (**J**) R_K, and (**K**) R_L.

**Figure 6 genes-12-00354-f006:**
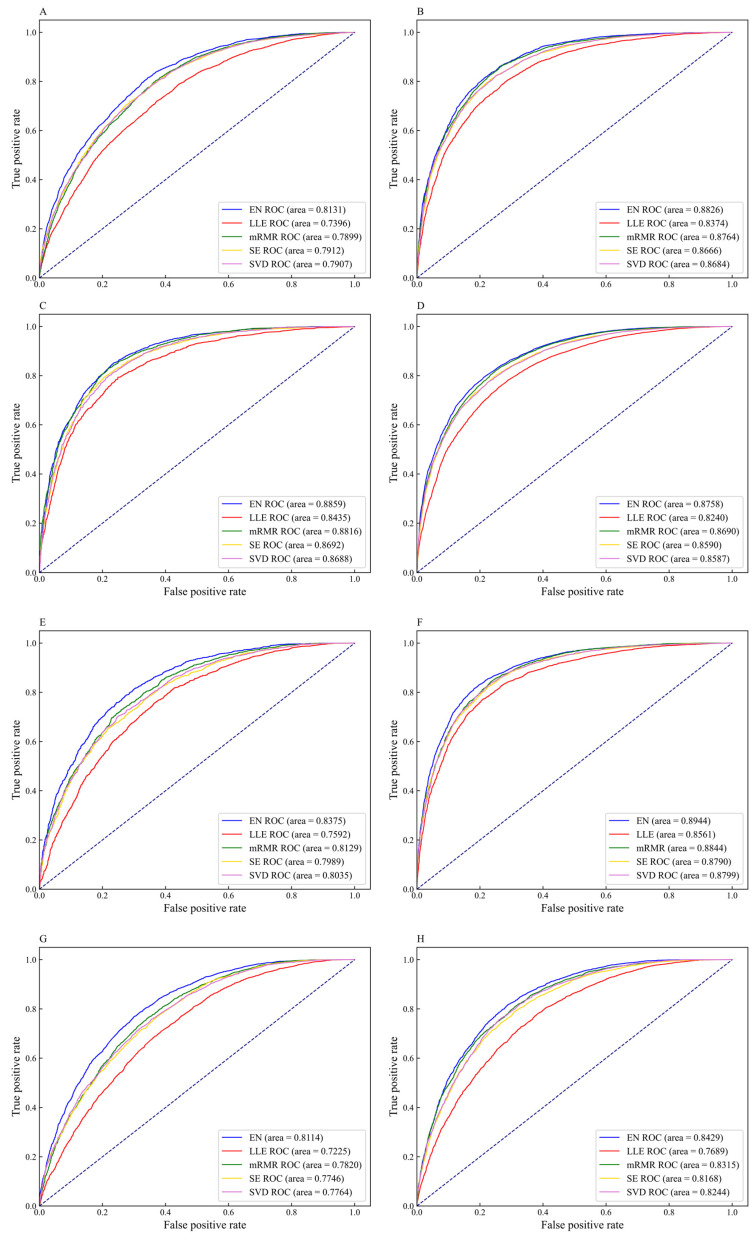
The receiver operator characteristic (ROC) curves of different feature selection methods on eleven datasets (**A**) H_B, (**B**) H_K, (**C**) H_L, (**D**) M_B, (**E**) M_H, (**F**) M_K, (**G**) M_L, (**H**) M_T, (**I**) R_B, (**J**) R_K, and (**K**) R_L.

**Figure 7 genes-12-00354-f007:**
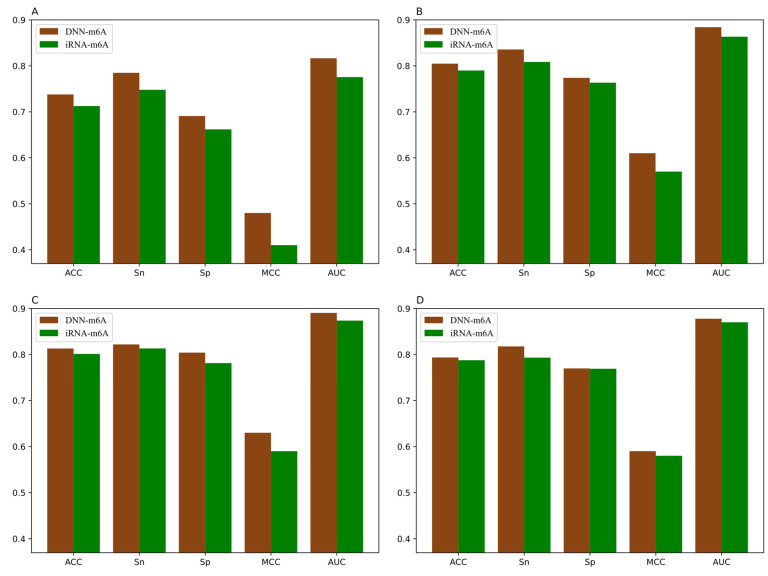
Performance of iRNA-m6A and DNN-m6A on eleven tissues’ training datasets (**A**) H_B, (**B**) H_K, (**C**) H_L, (**D**) M_B, (**E**) M_H, (**F**) M_K, (**G**) M_L, (**H**) M_T, (**I**) R_B, (**J**) R_K, and (**K**) R_L.

**Figure 8 genes-12-00354-f008:**
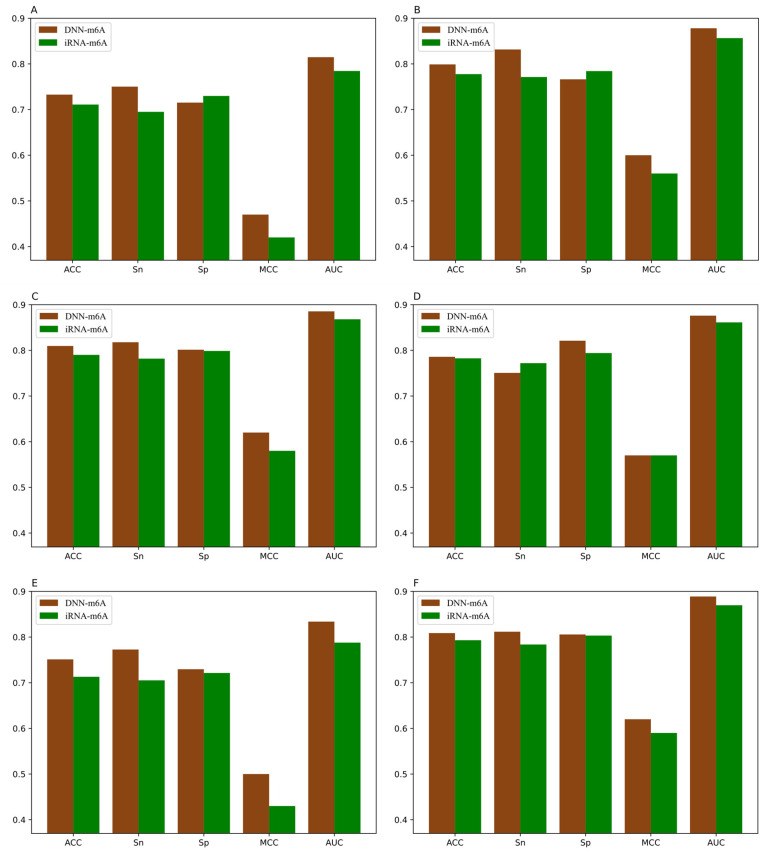
Performance of iRNA-m6A and DNN-m6A on eleven tissues’ independent datasets (**A**) H_B, (**B**) H_K, (**C**) H_L, (**D**) M_B, (**E**) M_H, (**F**) M_K, (**G**) M_L, (**H**) M_T, (**I**) R_B, (**J**) R_K, and (**K**) R_L.

**Figure 9 genes-12-00354-f009:**
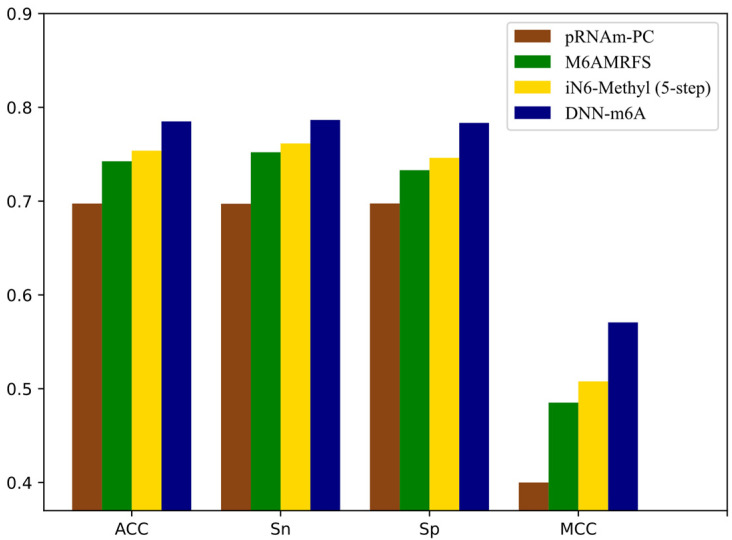
Comparison of pRNAm-PC, M6AMRFS, iN6-Methyl (5-step) and DNN-m6A on dataset S51.

**Table 1 genes-12-00354-t001:** The benchmark datasets for RNA m6A sites prediction.

Species	Tissues	Positive		Negative	
		Training	Testing	Training	Testing
Human	Brain	4605	4604	4605	4604
	Kidney	4574	4573	4574	4573
	Liver	2634	2634	2634	2634
Mouse	Brain	8025	8025	8025	8025
	Heart	2201	2200	2201	2200
	Kidney	3953	3952	3953	3952
	Liver	4133	4133	4133	4133
	Testis	4707	4706	4707	4706
Rat	Brain	2352	2351	2352	2351
	Kidney	3433	3432	3433	3432
	Liver	1762	1762	1762	1762

**Table 2 genes-12-00354-t002:** Chemical structure of each nucleotide.

Chemical Property	Class	Nucleotides
Ring Structure	Purine	A, G
	Pyrimidine	C, U
Functional Group	Amino	A, C
	Keto	G, U
Hydrogen Bond	Strong	C, G
	Weak	A, U

**Table 3 genes-12-00354-t003:** The ranges of hyper-parameters that need to be adjusted.

Hyper-Parameters	Meaning	Search Ranges
layers	number of hidden layers	(2,3)
hidden_1	number of neurons in the first hidden layer	(100, 800)
hidden_2	number of neurons in the second hidden layer	(50, 700)
hidden_3	number of neurons in the third hidden layer	(25, 600)
activation	activation function	elu, selu; softplus; softsign; relu; tanh; hard_sigmoid
optimizer	Per-parameter adaptive	RMSprop; Adam; Adamax; SGD; Nadam; Adadelta; Adagrad
learning_rate	learning rate of the optimizer	(0.001, 0.09)
kernel_initializer	layers weight initializer	uniform; normal; lecun_uniform; glorot_uniform; glorot_normal; he_normal; he_uniform
dropout	dropout rate	(0.1, 0.6)
epochs	number of iterations	10; 20; 30; 40; 50; 60; 70; 80; 90; 100
batch_size	number of samples for one training	40; 50; 60; 70; 80

**Table 4 genes-12-00354-t004:** The optimization results of the hyper-parameters.

Hyper- Parameters	H_B	H_K	H_L	M_B	M_H	M_K	M_L	M_T	R_B	R_K	R_L
layers	2	2	3	3	3	2	3	2	2	2	2
hidden_1	116	381	798	576	506	400	794	431	203	316	627
hidden_2	697	147	694	132	621	498	506	217	116	177	234
hidden_3	-	-	464	598	501	-	329	-	-	-	-
activation	softplus	selu	softsign	softplus	softsign	selu	selu	softplus	softplus	softplus	hard_sigmoid
optimizer	Adagrad	Adamax	Adadelta	Adagrad	Adadelta	Adagrad	Adagrad	SGD	Adamax	Adadelta	Adadelta
learning_rate	0.0373	0.0042	0.0441	0.0479	0.0587	0.0015	0.0026	0.0860	0.0027	0.0667	0.0899
kernel_initializer	glorot_normal	glorot_normal	lecun_uniform	lecun_uniform	uniform	uniform	lecun_uniform	glorot_uniform	he_uniform	he_normal	he_uniform
dropout	0.3233	0.4596	0.5073	0.2525	0.5971	0.4981	0.2401	0.4129	0.3226	0.1214	0.1840
epochs	70	10	100	20	70	30	50	80	20	50	30
batch_size	80	80	50	70	70	60	70	80	80	80	50

**Table 5 genes-12-00354-t005:** The performance of models before and after parameter optimization.

Species	Tissues	TPE	ACC	Sn	Sp	MCC	AUC
Human	Brain	Yes	**0.7378**	0.7848	0.6908	0.4788	0.8165
		No	0.7344	0.8165	0.6523	0.4764	0.8131
	Kidney	Yes	**0.8048**	0.8356	0.7739	0.6107	0.8841
		No	0.7984	0.8640	0.7328	0.6023	0.8826
	Liver	Yes	**0.8130**	0.8219	0.8041	0.6264	0.8905
		No	0.8077	0.8466	0.7688	0.6188	0.8859
Mouse	Brain	Yes	**0.7936**	0.8176	0.7697	0.5880	0.8778
		No	0.7890	0.8160	0.7621	0.5807	0.8758
	Heart	Yes	**0.7617**	0.7751	0.7483	0.5238	0.8439
		No	0.7565	0.7865	0.7265	0.5144	0.8375
	Kidney	Yes	**0.8196**	0.8320	0.8072	0.6396	0.8953
		No	0.8151	0.8530	0.7771	0.6331	0.8944
	Liver	Yes	**0.7358**	0.7757	0.6959	0.4733	0.8139
		No	0.7303	0.8210	0.6397	0.4697	0.8114
	Testis	Yes	**0.7662**	0.8099	0.7225	0.5347	0.8493
		No	0.7616	0.8007	0.7225	0.5259	0.8429
Rat	Brain	Yes	**0.7827**	0.7908	0.7746	0.5658	0.8678
		No	0.7819	0.8180	0.7457	0.5657	0.8672
	Kidney	Yes	**0.8338**	0.8427	0.8249	0.6679	0.9104
		No	0.8321	0.8488	0.8153	0.6658	0.9087
	Liver	Yes	**0.8263**	0.8417	0.8110	0.6533	0.8991
		No	0.8229	0.8428	0.8031	0.6474	0.8962

## Data Availability

Publicly available datasets were analyzed in this study. This data can be found here: http://lin-group.cn/server/iRNA-m6A/ (accessed on 27 February 2021).

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
