# Peer review of "DNN-m6A: A Cross-Species Method for Identifying RNA N6-methyladenosine Sites Based on Deep Neural Network with Multi-Information Fusion"

_genes, 2021, doi:10.3390/genes12030354_

Round 1

Reviewer 1 Report

In this study, the authors propose a computational method to identify m6A sites as an alternative to experimental methods. I have the following comments to improve the manuscript: 

  • In the abstract, there are qualitative remarks than need to be justified with quantification. For example, "superior to the state-of-the-art method" and "excellent generalization ability". At this point, we do not know whether you are referring to 100% accuracy or 90% accuracy (or both precision, recall). Please provide summary metrics in the abstract, if there is a claim of better performance.
  • The introduction covers methylation and its importance. It would be helpful briefly mentioning the why methylation occurs. It is mentioned that it is associated with diseases but how it is associated is missing. Does it always suppress gene expression or through other mechanisms impact the processes mentioned? 1-2 sentences on this would make the introduction more informative.
  • In results, tables 1-6 can be represented by a Figure such as heatmap, barplot (and the full tables can be given as supplemental) to improve readability.  Similarly tables 10-11 could be represented with a Figure.

Reviewer 2 Report

This manuscript describes a series of models for detecting m6a modifications in sequencing data. This modification is common, important and expensive to detect so improvements in methods of detection would be useful to the community.

Major comments

  • The manuscript requires English editing. For example the second sentence of the abstract reads awkwardly.

  • It is unclear how to run the code provided. The repository should provide clear instructions on how to fetch the example data sets and how to run the code. Ideally it would include a dockerfile that ensures the environment is properly setup as well.

  • The approach described may be overly specific. For example, the authors set the alpha value based upon the data source. However, in practice the data source may not be one of the provided examples. How will the method perform on a new data source? Has this approach produced a general model or a series of models that are only limited to the trained data sources?

Minor comments

  • This may be an issue with my pdf reader, but Table 8 appears to be cutoff.

Reviewer 3 Report

The authors developed a new method to predict RNA N6-methyladenosine sites. Their method uses a much larger number of RNA sequence features than the other methods. However, their method performed only slightly better than the other method, iRNA-m6A, which is disappointing. Nevertheless, prediction of RNA N6-methyladenosine sites is a very difficult task, and thus their method will be a good addition.

Major:

There are many computational methods for predicting RNA N6-methyladenosine sites, as they were described in the introduction (page 2). However, only one method (iRNA-m6A) was benchmarked to compare the performance with their method. Please clarify the reasons why the other methods were not used in the performance comparison.  

Round 2

Reviewer 1 Report

The authors have addressed all my comments and the manuscript was improved with the revisions. 

Reviewer 2 Report

I thank the authors for the changes to the manuscript. I believe they have clarified and improved the manuscript and I have no further comments.